# Karyotype Description and Comparative Chromosomal Mapping of 5S rDNA in 42 Species

**DOI:** 10.3390/genes15050647

**Published:** 2024-05-20

**Authors:** Xiaomei Luo, Yunke Liu, Xiao Gong, Meng Ye, Qiangang Xiao, Zhen Zeng

**Affiliations:** 1National Forestry and Grassland Administration Key Laboratory of Forest Resources Conservation and Ecological Safety on the Upper Reaches of the Yangtze River, College of Forestry, Sichuan Agricultural University, Chengdu 611130, China; gongxiao202209@163.com (X.G.); yemeng5581@163.com (M.Y.); 2Chengdu Academy of Agriculture and Forestry Sciences, Nongke Road 200, Wenjiang District, Chengdu 611130, China; lykchengdu@sina.com (Y.L.); xiaoqg1992@163.com (Q.X.); 15882215544@126.com (Z.Z.)

**Keywords:** chromosomes, conservation, diversity, FISH oligonucleotides probes

## Abstract

This study was conducted to evaluate the 5S rDNA site number, position, and origin of signal pattern diversity in 42 plant species using fluorescence in situ hybridization. The species were selected based on the discovery of karyotype rearrangement, or because 5S rDNA had not yet been explored the species. The chromosome number varied from 14 to 160, and the chromosome length ranged from 0.63 to 6.88 μm, with 21 species having small chromosomes (<3 μm). The chromosome numbers of three species and the 5S rDNA loci of nineteen species are reported for the first time. Six 5S rDNA signal pattern types were identified. The 5S rDNA varied and was abundant in signal site numbers (2–18), positions (distal, proximal, outside of chromosome arms), and even in signal intensity. Variation in the numbers and locations of 5S rDNA was observed in 20 species, whereas an extensive stable number and location of 5S rDNA was found in 22 species. The potential origin of the signal pattern diversity was proposed and discussed. These data characterized the variability of 5S rDNA within the karyotypes of the 42 species that exhibited chromosomal rearrangements and provided anchor points for genetic physical maps.

## 1. Introduction

Ribosomal DNA (rDNA) is essential to all cell types that code rRNA. rDNAs are detected by exercising considerable parts of chromosomes, including 45S and 5S rDNA [1]. There are numerous copies of 5S rDNA in genomes and this rDNA has been used in hundreds of cytogenetic investigations as an important marker for fluorescence in situ hybridization (FISH) analysis [2,3]. The 5S rDNA is not only for essential crops [4], but also for woody plants, such as walnuts [5], the Chinese pepper [6,7,8], and wintersweet [9].

The sequence lengths of 5S rDNA range from 48 bp [10] to 854 bp [11] according to the copies and variations available when searching for “5S rDNA” in the Nucleotide database of the National Center for Biotechnology Information (NCBI). Furthermore, the lengths of 5S rDNA as a FISH probe in the NCBI PubMed database varied considerably, at 41–1193 bp [12,13].

In previous research, the number of 5S rDNA FISH signal sites ranged from 1 to 71 [3]. Some species conserve 5S rDNA, including only two stable chromosomes with two 5S rDNA signal sites [14,15,16,17,18]. However, a few species occupy visible 5S rDNA FISH signals, including the numbers of both major and visible dispersed sites. There are four 5S rDNA signal sites in six in *Sinapidendron frutescens* (Aiton) Lowe [19], and up to 38 sites in *Paphiopedilum gigantifolium* Braem, M.L.Baker & C.O.Baker [20]. In addition, two 5S rDNA sites were found on the same chromosome of certain species, such as for *P. concolor* [12], *Brassica oleracea* L. [19], and *Prunus cerasus* L. [21].

Usually, the 5S rDNA FISH signal position is diverse. The 5S rDNA signal has been found in the chromosomal interstitial position of *Macroptilium bracteatum* (Nees & Mart.) Maréchal & Baudet [22], *Deschampsia antarctica* E. Desv [23], *Polemonium caeruleum* Linnaeus [24], in the chromosome distal position from *Pinus koraiensis* Siebold et Zuccarini [25], *Cannabis sativa* L. [16], *Pseudotsuga menziesii* (Mlrb.) Franco [26], in the chromosome proximal position for *Brassica rapa* L. [27], and even far from the chromosome, as reported for *Hedysarum setigerum* Turcz. [18] and *P. cerasus* [21].

Consequently, 5S rDNA as a FISH probe is an excellent marker for labeling plant chromosomes and distinguishing closely related species among more than 300 species (listed in Appendix A), including more than 20 woodies species (bold type in Appendix A). Examples of taxa distinguished using 5S rDNA as a FISH probe include six species of Fabaceae [12,28], five *Pinus* L. species of Pinaceae [25], four species of Oleaceae [29], two *Berberis* L. species of Berberidaceae [30], two species of Malvaceae [31,32], and *Zanthoxylum armatum* DC. of Rutaceae of [6,7]. However, the use of 5S rDNA as a FISH probe lacked discrimination (only stable two chromosomes with signal) in more than 40 species, including *Chimonanthus campanulatus* R.H. Chang & C.S. Ding [9], *Hippophaë rhamnoides* L. [33], *Juglans regia* L., and *Juglans sigillata* Dodeof [5].

In summary, 5S rDNA is a valuable cytogenetic marker owing to the tandem repeats and multiple copies with unusual chromosomal locations. These discoveries may provide a method for surveying the locations and number of rDNA diversifications among species and their respective relational accessions. Such findings will enhance understanding of the developmental and phylogenetic links of studied species. Patent oligonucleotide FISH (oligo-FISH) technology is claimed to be problematic in related cytogenetic investigations compared to conventional FISH analysis because of its low cost [34]. Moreover, oligonucleotide probes (oligo-probes) have been developed and successfully applied to many plants based on available DNA sequencing data [7,33,35,36,37]. This study aimed to analyze and compare the polymorphism of the signal patterns of 5S rDNA among 42 plant species, examining the 5S rDNA signal number, position, and intensity, and the chromosome number of each plant. Furthermore, several contributing factors caused by the diversity of the 5S rDNA signal pattern remain to be addressed.

## 2. Materials and Methods

The species used for this experiment were chosen due to the discovery of karyotype reshuffling [5,6,7,9,12,28,29,30,32,33,37,38,39]. In addition, the 5S rDNA of the species had not been explored previously. Information on the seeds or seedlings of these 64 plants (52 woody plants and 12 herbaceous plants belonging to 42 species, 30 genera, and 20 families) is provided in Table 1. All specimens of the 64 plants were collected from 23 counties or districts in six Chinese provinces.

### 2.1. Chromosome and Oligonucleotide Probe Preparation

The seeds of the analyzed species were germinated in Petri dishes with moistened filter paper. These seeds were maintained at 25 °C during the day and 18 °C at night until their roots reached approximately 2 cm in length. Subsequently, the roots of the germinated seeds were removed and the collected seedlings were grown in soil at room temperature (15–25 °C) until they produced new roots, which were cut again. The cut root tips were incubated in nitrous oxide (N_2_O) gas for 3–5 h, with a processing time according to cell wall lignification and chromosome size. The root tips were then soaked in glacial acetic acid for 5 min and in 75% ethyl alcohol afterward. Chromosome preparation was based on a study by Luo et al. [14]. Briefly, an enzymolysis procedure was performed at 1 mm of the meristematic zone of the root tip (cut root cap) at 37 °C by 45 min using pectinase and cellulase. There was a total of 50 mL (buffer was covered with 0.4324 g citric acid and 0.5707 g trisodium citrate, 1 mL buffer, 0.02 g pectinase, and 0.04 g cellulase). The enzymes were purchased from Kyowa Chemical Products Co., Ltd. (Osaka, Japan) and Yakult Pharmaceutical Ind. Co., Ltd. (Tokyo, Japan). The enzymes were then blended into suspension for dropping onto clean slides. These slides were air dried at room temperature and examined using an Olympus CX23 microscope (Olympus, Tokyo, Japan).

The oligoprobe of 5′ TCAGAACTCCGAAGTTAAGCGTGCTTGGGCGAGAGTAGTAC 3′ (41 bp) was initially observed in *P. nepalensis* (*P. concolor*, former name in *Flora of China*) of Fabaceae [12]. Subsequently, this oligoprobe was steadily executed in two *Berberis* species of Berberidaceae [30], *Chimonanthus campanulatus* of Calycanthaceae [9], cultural/wild *H. rhamnoides* ssp. *sinensis*, three *H. rhamnoides* cultivars of Elaeagnaceae [33], *A. fruticose* and *S. japonicum* of Fabaceae and three *Robinia* species [28], two *Juglans* species of Juglandaceae [5], *A. digitata* and *H. mutabilis* of Malvaceae [31,32]), *F. pennsylvanica* of Oleaceae, two species of *Ligustrum* and *S. oblata* [29], three *Bletilla* species of Orchidaceae [40], two *Zanthoxylum* species of Rutaceae [6,7], and five *Taxus* species of Taxaceae [38]. This oligoprobe was produced by Sangon Biotech Co., Ltd. (Shanghai, China) and conducted simultaneously in a single round of FISH. The oligoprobe was labeled with 6-carboxy-fluorescein (6-FAM; absorption/emission wavelengths 494 nm/518 nm; green).

### 2.2. FISH

Slides with well-spread chromosomes were used to hybridize the above oligo-probe. Chromosome samples were fixed (4% paraformaldehyde at room temperature, for 10 min), dehydration (75%, 95%, and 100% ethanol at room temperature for 5 min), denatured (deionized formamide at 80 °C for 2 min), and dehydrated again (75%, 95%, and 100% ethanol, at −20 °C for 5 min), Subsequently the chromosome samples were hybridized (0.375 μL of 5S rDNA, 4.675 μL of 2× SSC, and 4.95 μL of ddH_2_O in a total of 10 μL hybridization mixture) for 2 h in an incubator at 37 °C. The hybridized slides were then rinsed with 2× SSC and ddH_2_O twice for 5 min at room temperature and air dried before counterstaining with 4,6-diamidino-2-phenylindole (DAPI, Vector Laboratories, Inc., Burlingame, CA, USA) for 5 min according to the step described by Luo et al. [12]. Chromosomes were traced using an Olympus BX-63 microscope (Olympus Corporation, Tokyo, Japan), and FISH photographs were obtained using a DP-70 CCD camera allocated to the microscope.

### 2.3. Karyotype Analysis

Karyotype data were executed using Photoshop CC 2015 (Adobe Systems Inc., San Jose, CA, USA) and DP Manager (Olympus Corporation, Tokyo, Japan). More than eight slides of each plant were observed, and at least 15 well-spread cells were selected to determine the chromosome number and length. All examined chromosomes were assembled from longest to shortest. The chromosome ratio was controlled by the length of the longest chromosome to that of the shortest chromosome. Exhaustive and deep karyotype analysis could not be conducted because of the obscure centromere position and small chromosome size of many of the analyzed species.

## 3. Results

### 3.1. Karyotype Analysis Revealed Differences among 42 Species

We performed a FISH analysis to visualize the chromosomal distribution of 5S rDNA in 42 plants species, as shown in Figure 1 (A1–A16), Figure 2 (B1–B16), Figure 3 (C1–C16) and Figure 4 (D1–D16). We cut each chromosome distribution of the 5S rDNA in Figure 1, Figure 2, Figure 3 and Figure 4 and aligned them for display, as shown in Appendix A. This is the first time that 5S rDNA testing has been performed and analyzed for 19 species from 13 families (A1, A8–A11, A13–A16, B11, B14, C12, C14–C16, D6–D12, and D14–D15).

The chromosome numbers and lengths for the examined species are sorted in Table 2. The chromosome numbers in the 42 species ranged from 14 (*C. chinensis*, A1) to 160 (*Z. bungeanum* ‘Hanyuanhuajiao’ 3, D11). Eight species had 24 chromosomes (19%), whereas five species (12%) had 34 chromosomes. The chromosome number in three species was determined for the first time (asterisk in Table 2): *P. sibiricum* (2n = 18), *I. chinensis* (2n = 40), and *T. sebifera* (2n = 88). The longest chromosome length of each plant ranged from 1.23 μm (*T. sebifera*, D6) to 6.88 μm (*T.* × *media*, B2; *T. aestivum*, D1), whereas the shortest chromosome length of each plant ranged from 0.63 μm (*J. sigillata* ‘Maerkang’ B16) to 3.85 μm (*T. aestivum*, D1). A total of 21 species (50%) had a chromosome length of less than 3 μm, thus were in the small chromosome category. Detailed and deep karyotype analysis was not conducted because of the unclear position of centromeres and the chromosomes in many of the examined plants, such as long/short arm length and karyotype formula.

Karyotype asymmetry (Table 2) was assessed using the longest to shortest chromosome length ratio. The largest ratio was 4.61 for *P. sibiricum* (A10), whereas the smallest was 0.74 for *R. pseudoacacia* ‘Idaho’ (A5). The ratio for 22 species ranged from 2 to 3 (52%), while that for 11 species ranged from 1 to 2 (26%), and seven plants ranged from 3 to 4 (17%). The ratio was >4 for two species—*P. sibiricum* (A10) and *B. striata* f. ‘Dujiangyan’ (C4)—whereas the ratio was <1 for *R. pseudoacacia* ‘Idaho’ (A5). These results indicate various differences in the karyotypes of the 42 examined in this study.

### 3.2. Diverse Signal Patterns of 5S rDNA Reveal a Complex Genome Architecture

To further investigate the diversity of 5S rDNA, different types of ideograms for the 42 species were drawn (Appendix A) based on the FISH karyograms shown in Appendix A. The first diversity of 5S rDNA was the signal location. Proximal signals were observed at several chromosomes in 31 species (74%), while distal signals were observed at several chromosome terminus in 28 species (67%). Interstitial signals were observed at several chromosomes in 19 species (45%), whereas distal signals deviated from the chromosome in four species (10%, the fourth class): *C. chinensis* (A1), *S. japonicum* (A6), *K. paniculata* (B14), and *Z. nitidum* (D7). The second diversity was the signal number. The largest number of chromosomes with a 5S rDNA signal was 18 in *T. sebifera* (D6), while the smallest number was two in 22 species (52%). Seven species (17%) contained 10–16 chromosomes with a 5S rDNA signal and 18 species (43%) had 4–8 chromosomes with a 5S rDNA signal. The ratio of chromosomes with a 5S rDNA signal to the total chromosome was assessed to signal cover. The largest ratio was 0.89 in *P. nepalensis* (A2), and the smallest ratio was 0.03 in *Z. nitidum* (D7) and *Z. armatum* ‘Putaoqingjiao’ (D16). The ratio for 31 species (74%) ranged from 0.03 to 0.20, while the ratio for eight species (19%) ranged from 0.20 to 0.50. Only two species had ratios ranging from 0.50 to 0.89: *R. pseudoacacia* (A3) and *T. wallichiana* var. *mairei* (B5).

We summarized the results in Appendix A to produce the 5S rDNA signal pattern in Figure 5. The results for 42 species belonging to 20 families are shown in different colors. There were six 5S rDNA signal pattern types: type I, chromosome includes the proximal signal location; type II, chromosome consists of the distal signal location; type III, chromosome consists of the proximal and distal signal locations; type IV, chromosome only consists of signal outside of the chromosome; type V, chromosome consists of the distal signal location and signal outside of the chromosome; and type VI, satellite chromosome consists of distal signal location. These types of signal patterns indicate that there is various diversity in the 5S rDNA signal arrangement.

In the 42 species examined in this study, 10 signal pattern types or type combinations were present (Figure 5). Twenty six plants only possessed signal pattern type I; nine plants only possessed signal pattern type II; sixteen plants had a combination of type I + type II; six plants only had type III; *R. pseudoacacia* f. *decaisneana* had a combination of type I + type III; *T. wallichiana* var. *mairei* had a combination of type I + type II + type III; S. japonicum only had type IV; *C. chinensis* and *K. bipinnata* had a combination of type II + type IV; *Z. nitidum* only had type V; and *P. nepalensis* had a combination of type I + type II + type III + type VI.

There were diverse signal patterns of 5S rDNA among the 42 species, indicating a complex genome architecture. In the family Rutaceae, four varieties of *Zanthoxylum* had type I, but five varieties of *Zanthoxylum* had a combination of type I + type II, and *Z. nitidum* had type V. In Fabaceae, *A. fruticose* and *E. crista-galli* had type I, but *R. pseudoacacia* and *R. pseudoacacia* ‘Idaho’ had type III, *R. pseudoacacia* f. *decaisneana* had combination of type I + type III, *S. japonicum* only had type IV, *C. chinensis* had a combination of type II + type IV, and *P. nepalensis* had a combination of type I + type II + type III + type VI. In the family Taxaceae, four species of Taxus had type III, but *T. wallichiana* var. *mairei* had a combination of type I + type II + type III. In Oleaceae, *F. pennsylvanica* had type II, but *L. lucidum*, *L.* × *vicaryi* and *S. oblata* had a combination of type I + type III. In Berberidaceae, *B. diaphana* had type II, but *B. soulieana* had a combination of type I + type II. In the family Malvaceae, *F. simplex* had type I, but *H. mutabilis* had a combination of type I + type II. In Lauraceae, *L. baviensis* had type I, but *L. baviensis* had a combination of type I + type II.

A summarized 5S rDNA signal diversity is illustrated in Figure 6. There were three major groups contributing to this diversity: (i) signal number: increases or decreases; (ii) signal location: on distal chromosome or proximal chromosome, or deviation from the chromosome; (iii) signal size: increases or decreases, or the normal signal size.

## 4. Discussion

### 4.1. Karyotype Analysis of the 42 Species

Traditional karyotype analysis involves counting chromosome numbers, determining centromere location, and measuring chromosome length and long/short arm ratio. Based on this, the karyotype formula and cytotype are obtained to compare whether the species is evolving [41]. The chromosome numbers of the 42 species in this study ranged from 2n = 14 to 160. Most chromosome numbers were consistent with previous studies [12,28,29,30,38,39,42,43,44,45,46], with only nine species having different chromosome numbers. In addition, the chromosome number of three species was analyzed for the first time in this study: *P. sibiricum* (2n = 18), *I. chinensis* (2n = 40), and *T. sebifera* (2n = 88). More information related to *T. sebifera* was found in the chromosome numbers of a previous research study. A stable chromosome number was found in the genus *Ilex* L.: *Ilex crenata* Thunb. (2n = 40), *I. makinoi* ‘Hara’ (2n = 40), *I. leucoclada* (2n = 40), and *I. yunnanensis* var. *gentilis* Franch. (2n = 160) [47]. Conversely, chromosome numbers varied in four genera: (i) *Polygonatum anhuiense* D. C. Zhang et J. Z. Shao (2n = 24), *Polygonatum langyaense* D. C. Zhang et J. Z. Shao (2n = 18), *Polygonatum odoratum* (Mill.) Druce (2n = 18), *Polygonatum zanlanscianense* Pamp. (2n = 28), *P. cyrtonema* (2n = 22) [43,44]; (ii) *Zanthoxylum acanthopodium* Candelle (2n = 64), *Zanthoxylum dimorphophyllum* Hemsley (2n = 36/68), *Zanthoxylum scandens* Blume (2n = 68), *Zanthoxylum oxyphyllum* Edgeworth (2n = 72), *Zanthoxylum tomentellum* J.D. Hance (2n = 72), *Zanthoxylum simulans* Hance (2n = ~132, *Z. nitidum* (2n = 68), *Z. armatum* (2n = 66/98/128/132/136), *Z. bungeanum* (2n = 132/136) [6,7,8,37,48,49,50]; (iii) *Bletilla formosana* (2n = 32/36), *B. striata* (2n = 32/34/36/48/51/64/76), *B. ochracea* (2n = 34/36) [39,40,46,51]; (iv) *J. regia* and *J. sigillata* (2n = 34) [5], *Juglans* (2n = 32) [52,53,54,55,56]. However, in this study, the results for *P. cyrtonema* (2n = 18), *Z. bungeanum* (2n = 76/134/136/160), *Z. armatum* (2n = 96/100/102/104/132), *Z. nitidum* (2n = 66), *B. formosana*, *B. striata*, *B. ochracea*, *J. regia*, and *J. sigillata* (all 2n = 34) contradicted those of previous studies. The possible causes of inconsistency may be related to (i) improper chromosome count in the small and high chromosomes, (ii) root lignification limiting their chromosome preparation of chromosomes from these species, (iii) hybridization between closely related species, (iv) natural or artificial polyploidization, and (v) apomixis (polyembryo).

Intraspecific chromosome number variation, even in the entire population, has also been found in species such as *Cuscuta epithymum* (L.) L. and *Cuscuta planiflora* Ten. [57], where most variation was attributable to auto- or allopolyploidy. The additional numbers can be explained by ascending or descending dysploidy. Thus, the accumulation of repetitive DNA can lead to an increase in chromosomes and, consequently, to an increase in genome size, especially in subgenus *Monogynella* [58]. In our study, chromosome numbers varied in the interspecific and intraspecific populations of the genus *Zanthoxylum*. The cause of the variation was probably similar to that of *Cuscuta* and *Monogynella*. Furthermore, the stable differentiation in the 5S rDNA FISH pattern between the subgenera suggests that chromosomal rearrangements played a role in splitting the two subgenera. Rather than major structural changes, transpositional events are responsible for the variable rDNA distribution patterns among species of the same subgenus with conserved karyotypes [25]. *Zanthoxylum* genomes have complex chromosome rearrangements, such as chromosomal fission, reversal, and translocations [8], which also explains the chromosome number variation in this genus. Chromosome polymorphisms within species in natural populations of vertebrates are far less common and are believed to be temporary transitions during chromosomal evolution [59,60]. Likewise, the genus *Zanthoxylum* may be experiencing chromosomal evolution.

The longest chromosome length of the 64 plants evaluated in this study ranged from 1.23 to 6.88 μm. In contrast, the shortest chromosome length of each plant ranged from 0.63 to 3.85 μm, exhibiting striking differences among the examined species. Previous research has shown the accumulated chromosome lengths of hundreds of plant species [6,9,12,28,30,32,33,37,38,61,62,63]. Analyzing these data, it is not difficult to find slight differences in chromosome length, even for the same accession of the same species. For example, *R. pseudoacacia* has reported chromosome lengths of 1.12–1.74 μm [37] and 0.94–1.67 μm [28]. Nonetheless, these two chromosome lengths were small (<3 μm). Hence, chromosome length was more suitable for qualitative rather than for quantitative analyses. Thirty-seven plant species analyzed (more than half) had chromosome lengths < 3 μm, placing them in the small chromosome rank. Owing to the hazy centromere mark and tiny chromosomes in many of the investigated plants, the chromosome size was determined by metaphase and the measurement method. A more delicate karyotype analysis (e.g., arm length, karyotype, and cytotype) was unavailable and limited.

### 4.2. Occurrence and Diversity of 5S rDNA in Plants

The 5S rDNA, which occurs in all cellular life forms, is a highly stable tandem repeat sequence that ubiquitously exists in plants [64]. With the evolution and development of the plant, 5S rDNA also underwent simultaneous changes. The length of 5S rDNA in the NCBI Nucleotide database ranged from 48 to 854 bp [10,11], while its length as a FISH probe in the PubMed database of NCBI ranged from 41 to 1193 bp [12,13,31]. This study was the first time that the 41-bp oligoprobe had been used to analyze 5S rDNA for 19 species from 13 families. Overall, 5S rDNA occurred in at least two chromosomes in all 42 species. With advances in science and technology, 5S rDNA has been confirmed in an increasing number of species [3,22,65,66]. However, whether the reported length of 5S rDNA is a complete or partial sequence, there are no doubt considerable differences among these 5S rDNAs, including the length and base pair [17,67,68].

Accordingly, it is reasonable that the chromosomally diverse distribution of 5S rDNA is visualized by FISH. Previous studies have shown that the numbers of 5S rDNA FISH signal sites range from 1 to 71 [3,12,19,20,69]. Such a signal position was found in the chromosome interstitial position, distal position, proximal position, and far away from the chromosome [3,21,25,26,27]. In this study, 5S rDNA was diverse and abundant in signal site number (2–18), position (e.g., interstitial, distal, proximal position, and occasionally, outside the chromosome), and even in intensity (e.g., strong, weak, slight). These findings are consistent with previous studies of the 5S rDNA signal pattern of *A. fruticose*, *B. formosana* ‘Leshan’, *C. campanulatus*, *H. mutabilis*, *P. nepalensis*, *S. oblata*, two species of *Berberis*, two varieties of *J. regia*, two varieties of *J. sigillata*, two species of *Ligustrum*, two varieties of *Robinia*, and two varieties of *Z. armatum*, wild/cultural *H. rhamnoides* ssp. *sinensis*, three varieties of *H. rhamnoides*, four species of *Taxus*, and six types of *B. striata* [6,9,12,29,30,33,37]. In contrast, the 5S rDNA signal pattern determined in this study differed from those previously reported for *F. pennsylvanica*, *R. pseudoacacia*, *S. japonicum*, *T. wallichiana* var. *mairei*, *Z. bungeanum* ‘Hanyuanhuajiao’, two types of *B. ochracea*, and two varieties of *Z. armatum* [7,29,40,44]. Possible causes of the 5S rDNA signal pattern discrepancy are lost satellite chromosomes with signals, and variation in different batches of materials in the same species (i.e., intraspecific variation).

### 4.3. Potential Origin of 5S rDNA Diversity in Plants

The diversity of signal patterns means 5S rDNA has been used as an excellent and dynamic marker in the species of *F. pennsylvanica*, *Iris versicolor* L., *L. vicaryi*, *L. lucidum*, *P. nepalensis* (*P. concolor*, former name in Flora of China), *R. pseudoacacia*, and *S. oblata* [12,28,29,70]. After comparing the 5S rDNA obtained in previous studies with ours, both perfectly reflect the extensive diversity in *P. nepalensis*, in which all 18 chromosomes can be distinguished according to the signal position, signal intensity, and signal number. Nevertheless, 5S rDNA was quite conserved and dormant in numerous species, such as *C. campanulatus*, *C. sativa*, *H. rhamnoides*, *J. regia*, *M. atropurpureum*, *P. stratiotes*, *P. trichocarpa*, and *Sanguisorba* L. [9,16,22,33,36,71,72]; leaving a question about the big difference in 5S rDNA among the above species. There are several plausible hypotheses under investigation, such as (i) chromosome rearrangement (e.g., deletion, duplication, inversion, translocation), (ii) polyploidization, (iii) self-incompatibility, and (iv) chromosome satellites.

The 5S rDNA position is a hot topic for chromosomal reshuffling because of its system into long reaches of the standpat tandem repetition unit and its active transcription. This feature implies that they are impressionable to chromosomal destruction or non-allelic homologous recombination, thus raising the feasibility of chromosomal reorganization, such as fissions, inversions, and fusions [73,74,75,76,77]. The 5S rDNA diversification was regarded as variable genomic areas, compliant with double-strand break and chromosomal reorganization, facilitating karyotypic reconstruction [77,78,79]. The 5S rDNA position was also diversified by translocation or transposition events of repeats in those chromosomes [80]. An interstitial 5S rDNA position with a diverse location is presumed to indicate a thin inversion. A plus 5S rDNA implies the presence of a replication [81]. In addition, the 5S rDNA site of one parent was either excluded from the chromosome or shifted into gene silencing and then disappeared, which could decrease the 5S rDNA site number [82,83]. These studies demonstrate that chromosome rearrangement causes variations in 5S rDNA diversity.

The 5S rDNA has a polyploidization-relevant preference to the distal from a proximal position but keeps a stable loci number [14]. The 5S rDNA sites are proximal, a highly transparent direction in chromosomes with a single site [84]. Consequently, the presence of the 5S rDNA site in the distal chromosomes and the abundance of microsatellites in adjacent areas provide conducive conditions for additional reshuffling. These results emphasize the effect of variable chromosomal 5S rDNA loci in generating assignments [85]. There are no associations between the number of 5S loci and chromosome number, but there is correspondence with ploidy level and genome size [20,84,86], although anomalies still occur [64,87,88], such as the genus *Cuscuta* L. [84]. These studies demonstrate that polyploidization causes variation in 5S rDNA diversity.

In summary, the variations 5S rDNA in signal number, location, and size were caused by chromosome rearrangement (e.g., deletion, duplication, inversion, translocation), recombination, self-incompatibility, as well as chromosome satellites.

## 5. Conclusions

In this study, we conducted FISH-based chromosomal mapping of 5S rDNA markers to provide valid karyotype landmarks for revealing the chromosome number and 5S rDNA signal pattern distribution in 42 plant species. Furthermore, we established chromosome physical mapping of each species. Finally, we discussed the proposed origin of 5S rDNA diversity in plants. We are devoted to developing universal oligo sequence markers (GAA)_6_, (TTG)_6_, (ACT)_6_, 45S, ITS, and combinatory analysis of additional plant species, particularly woody plants. Collectively, our results enhance the assumption that cytogenetic characteristics (conventional and molecular) are excellent markers for chromosome distinction and the presentation and profile of existing biodiversity in woody plants.

## Figures and Tables

**Figure 1 genes-15-00647-f001:**
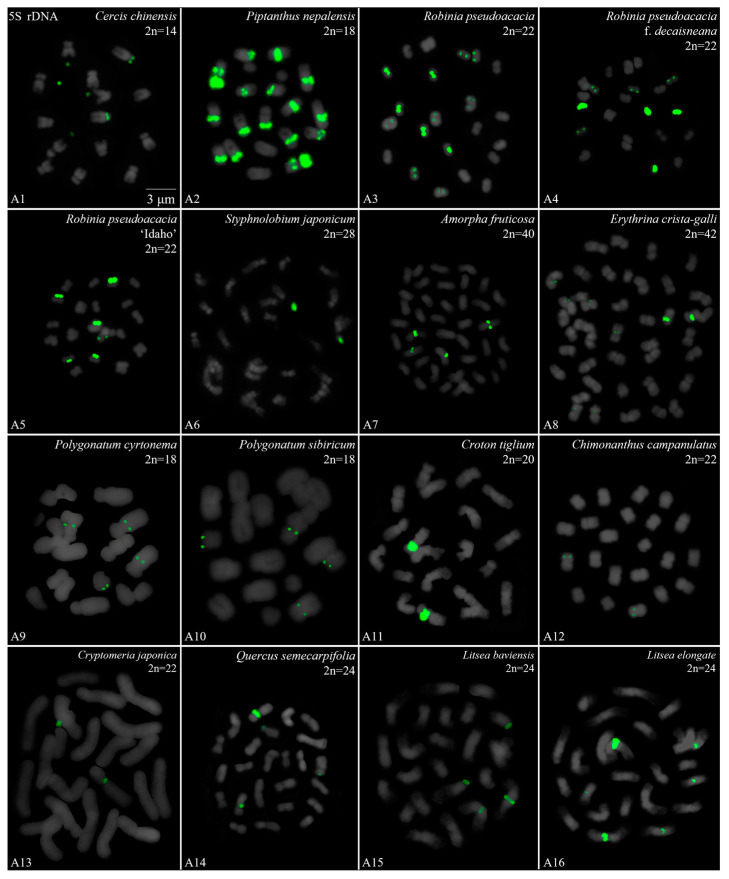
Oligo-FISH depicting the 5S rDNA present in 16 plants. Chromosomes in (**A1**–**A16**) are all from metaphase. Oligo-probe 5S rDNA is exhibited by green signals: (**A1**) *C. chinensis*, 2n = 14; (**A2**) *P. nepalensis*, 2n = 18; (**A3**) *R. pseudoacacia*, 2n = 22; (**A4**) *R. pseudoacacia* f. *decaisneana*, 2n = 22; (**A5**) *R. pseudoacacia* ‘Idaho’, 2n = 22; (**A6**) *S. japonicum*, 2n = 22; (**A7**) *A. fruticosa*, 2n = 22; (**A8**) *E. crista-galli*, 2n = 42; (**A9**) *P. cyrtonema*, 2n = 18; (**A10**) *P. sibiricum*, 2n = 18; (**A11**) *C. tiglium*, 2n = 20; (**A12**) *C. campanulatus*, 2n = 22; (**A13**) *C. japonica*, 2n = 22; (**A14**) *Q. semecarpifolia*, 2n = 24; (**A15**) *L. baviensis*, 2n = 24; (**A16**) *L. elongate*, 2n = 24. Bar: 3 μm. (**A1**,**A8**–**A11**,**A13**–**A16**) these are the first time that 5S rDNA testing has been reported.

**Figure 2 genes-15-00647-f002:**
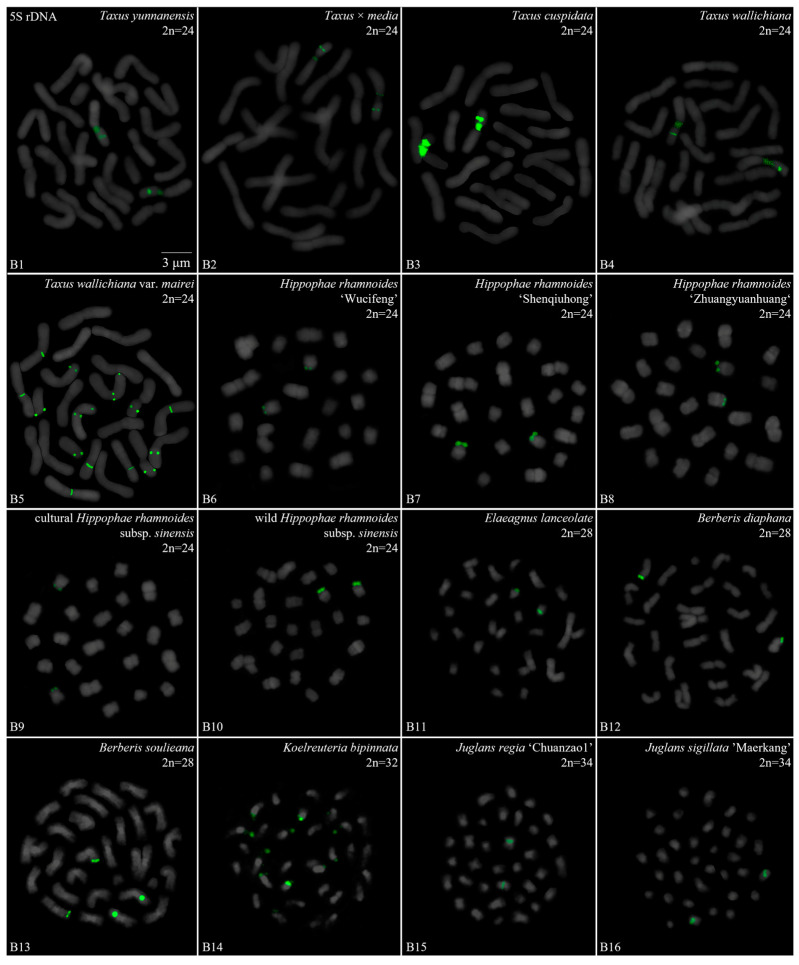
Oligo-FISH depicting the 5S rDNA present in 16 plants. Chromosomes in (**B1**–**B16**) are all from metaphase. Oligo-probe 5S rDNA is exhibited by green signals: (**B1**) *T. yunnanensis*, 2n = 24; (**B2**) *T. media*, 2n = 24; (**B3**) *T. cuspidata*, 2n = 24; (**B4**) *T. wallichiana*, 2n = 24; (**B5**) *T. wallichiana* var. *mairei*, 2n = 24; (**B6**) *H. rhamnoides* ‘Wucifeng’, 2n = 24; (**B7**) *H. rhamnoides* ‘Shengqiuhong’, 2n = 24; (**B8**) *H. rhamnoides* ‘Zhuangyuanhuang’, 2n = 24; (**B9**) cultural *H. rhamnoides* ssp. *sinensis*, 2n = 24; (**B10**) wild *H. rhamnoides* ssp. *sinensis*, 2n = 24; (**B11**) *E. lanceolata*, 2n = 28; (**B12**) *B. diaphana*, 2n = 28; (**B13**) *B. soulieana*, 2n = 28; (**B14**) *K. paniculata*, 2n = 32; (**B15**) *J. regia* ‘Chuanzao1’, 2n = 34; (**B16**) *J. sigillata* ‘Maerkang’, 2n = 34. Bar: 3 μm. (**B11**,**B14**) these are the first time that 5S rDNA testing has been reported.

**Figure 3 genes-15-00647-f003:**
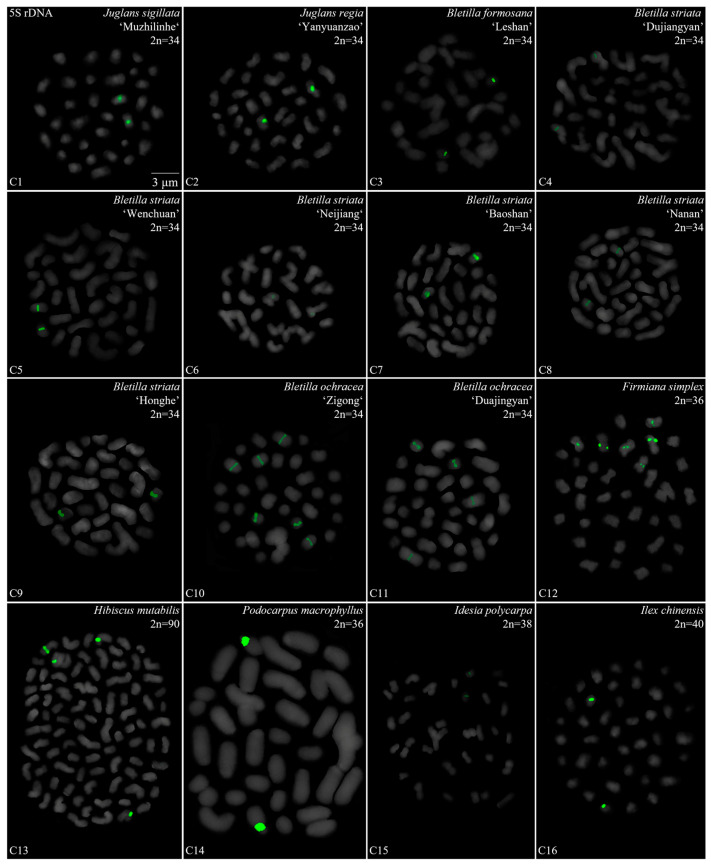
Oligo-FISH depicting the 5S rDNA present in 16 plants. Chromosomes in (**C1**–**C16**) are all from metaphase. Oligo-probe 5S rDNA is exhibited by green signals: (**C1**) *J. sigillata* ‘Muzhilinhe’, 2n = 34; (**C2**) *J. regia* ‘Yanyuanzao’, 2n = 34; (**C3**) *B. formosana* ‘Leshan’, 2n = 34; (**C4**) *B. striata* ‘Dujiangyan’, 2n = 34; (**C5**) *B. striata* ‘Wenchuan’, 2n = 34; (**C6**) *B. striata* ‘Neijiang’, 2n = 34; (**C7**) *B. striata* ‘Baoshan’, 2n = 34; (**C8**) *B. striata* ‘Nanan’, 2n = 34; (**C9**) *B. striata* ‘Honghe’, 2n = 34; (**C10**) *B. ochracea* ‘Zigong’, 2n = 34; (**C11**) *B. ochracea* ‘Dujiangyan’, 2n = 34; (**C12**) *F. simplex*, 2n = 36; (**C13**) *H. mutabilis*, 2n = 90; (**C14**) *P. macrophyllus*, 2n = 36; (**C15**) *I. polycarpa*, 2n = 38; (**C16**) *I. chinensis*. 2n = 40. Bar: 3 μm. (**C12**,**C14**–**C16**) these are the first time that 5S rDNA testing has been reported.

**Figure 4 genes-15-00647-f004:**
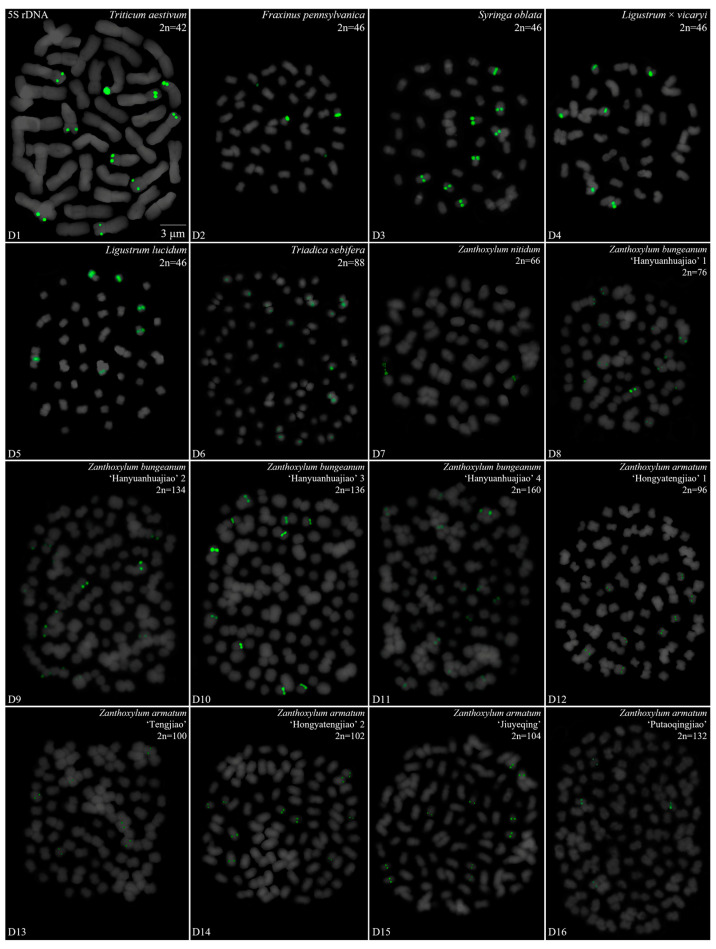
Oligo-FISH depicting the 5S rDNA present in 16 plants. Chromosomes in (**D1**–**D16**) are all from metaphase. Oligo-probe 5S rDNA is exhibited by green signals: (**D1**) *T. aestivum*, 2n = 42; (**D2**) *F. pennsylvanica*, 2n = 46; (**D3**) *S. oblata*, 2n = 46; (**D4**) *L.* × *vicaryi*, 2n = 46; (**D5**) *L. lucidum*, 2n = 46; (**D6**) *T. sebifera*, 2n = 88; (**D7**) *Z. nitidum*, 2n = 66; (**D8**) *Z. bungeanum* ‘Hanyuanhuajiao’ 4, 2n = 76; (**D9**) *Z. bungeanum* ‘Hanyuanhuajiao’ 1, 2n = 134; (**D10**) *Z. bungeanum* ‘Hanyuanhuajiao’ 2, 2n = 136; (**D11**) *Z. bungeanum* ‘Hanyuanhuajiao’ 3, 2n = 160; (**D12**) *Z. armatum* ‘Hongyatengjiao’ 1, 2n = 96; (**D13**) *Z. armatum* ‘Tengjiao’, 2n = 100; (**D14**) *Z. armatum* ‘Hongyatengjiao’ 2, 2n = 102; (**D15**) *Z. armatum* ‘Jiuyeqing’, 2n = 104; (**D16**) *Z. armatum* ‘Putaoqingjiao’, 2n = 132. Bar: 3 μm. (**D6**–**D12**,**D14**,**D15**) these are the first time that 5S rDNA testing has been reported.

**Figure 5 genes-15-00647-f005:**
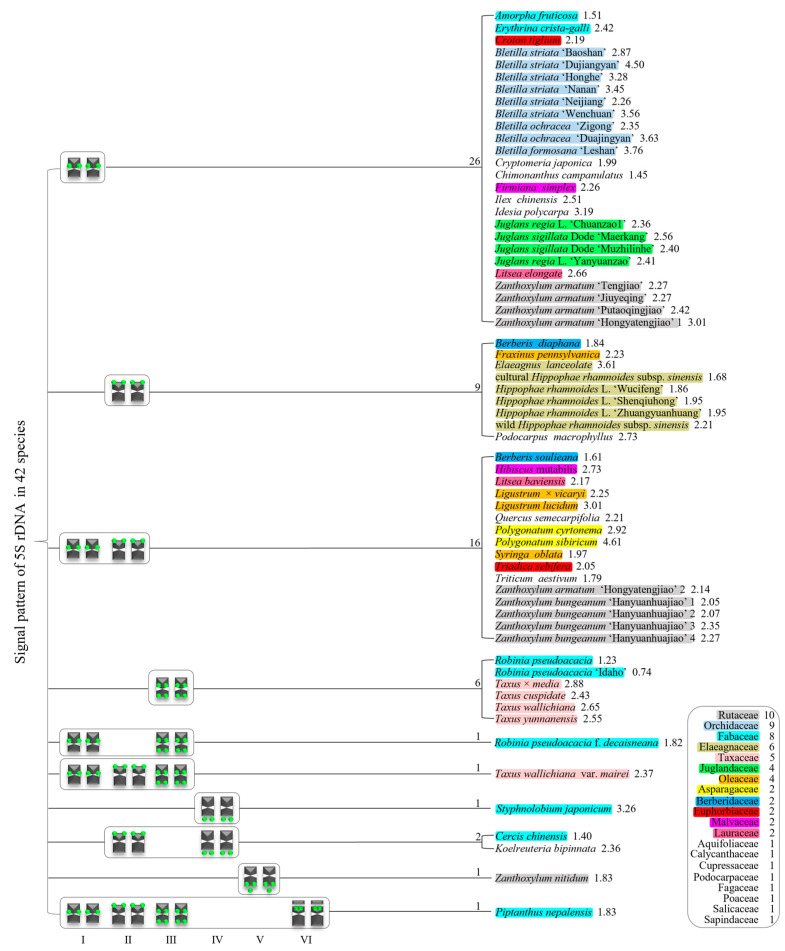
Signal pattern of 5S rDNA in 42 species. Signal patterns type I–VI are summarized based on Appendix A. The number between the signal pattern and the species represents the species number. The number after the species represents the ratio of longest to shortest chromosome length. Rutaceae includes 10 plants (grey); Orchidaceae includes nine plants (light blue); Fabaceae includes eight plants (cyan); Elaeagnaceae includes six plants (light yellow); Taxaceae includes five plants (light pink); Juglandaceae and Oleaceae both include four plants (green and orange); Asparagaceae, Berberidaceae, Euphorbiaceae, Malvaceae, and Lauraceae all include two plants (yellow, blue, red, magenta, pink); Aquifoliaceae, Calycanthaceae, Cupressaceae, Podocarpaceae, Fagaceae, Poaceae, Salicaceae, and Sapindaceae each include one plant, respectively.

**Figure 6 genes-15-00647-f006:**
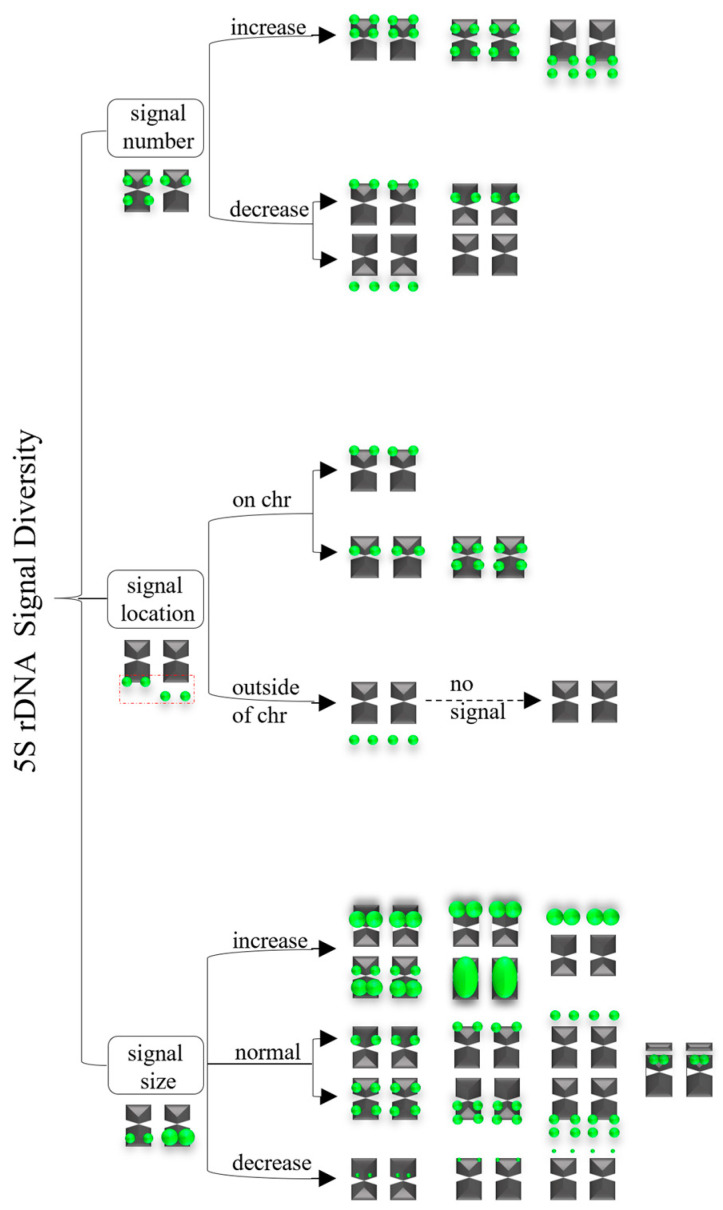
The 5S rDNA signal diversity.

**Table 1 genes-15-00647-t001:** Details of all 64 plants used in this work.

Family	No.	Species	Collection Location
Rutaceae	1	*Zanthoxylum nitidum* (Roxb.) DC.	Hanyuan, Sichuan
	2	*Zanthoxylum bungeanum* Maxim. ‘Hanyuanhuajiao’ 1	Hanyuan, Sichuan
	3	*Zanthoxylum bungeanum* Maxim. ‘Hanyuanhuajiao’ 2	Hanyuan, Sichuan
	4	*Zanthoxylum bungeanum* Maxim. ‘Hanyuanhuajiao’ 3	Hanyuan, Sichuan
	5	*Zanthoxylum bungeanum* Maxim. ‘Hanyuanhuajiao’ 4	Hanyuan, Sichuan
	6	*Zanthoxylum armatum* DC. ‘Hongyatengjiao’ 1	Hongya, Sichuan
	7	*Zanthoxylum armatum* DC. ‘Hongyatengjiao’ 2	Hongya, Sichuan
	8	*Zanthoxylum armatum* DC. ‘Jiuyeqing’	Jiangjin, Chongqing
	9	*Zanthoxylum armatum* DC. ‘Putaoqingjiao’	Hanyuan, Sichuan
	10	*Zanthoxylum armatum* DC. ‘Tengjiao’	Hanyuan, Sichuan
Orchidaceae	11	*Bletilla striata* (Thunb. ex Murray) Rchb. f. ‘Dujiangyan’	Dujiangyan, Sichuan
	12	*Bletilla striata* (Thunb. ex Murray) Rchb. f. ‘Neijiang’	Neijiang, Sichuan
	13	*Bletilla striata* (Thunb. ex Murray) Rchb. f. ‘Wenchuan’	Wenchuan, Sichuan
	14	*Bletilla striata* (Thunb. ex Murray) Rchb. f. ‘Baoshan’	Baoshan, Yunnan
	15	*Bletilla striata* (Thunb. ex Murray) Rchb. f. ‘Honghe’	Honghe, Yunnan
	16	*Bletilla striata* (Thunb. ex Murray) Rchb. f. ‘Nanan’	Nanan, Chongqing
	17	*Bletilla ochracea* Schltr. ‘Dujiangyan’	Dujiangyan, Sichuan
	18	*Bletilla ochracea* Schltr. ‘Zigong’	Zigong, Sichuan
	19	*Bletilla formosana* (Hayata) Schltr. ‘Leshan’	Leshan, Sichuan
Fabaceae	20	*Cercis chinensis* Bunge	Wenjiang, Sichuan
	21	*Piptanthus nepalensis* (Hook.) D. Don	Qingbaijiang, Sichuan
	22	*Robinia pseudoacacia* L.	Wenjiang, Sichuan
	23	*Robinia pseudoacacia* f. *decaisneana* (Carr.) Voss	Suqian, Jiangsu
	24	*Robinia pseudoacacia* ‘Idaho’	Suqian, Jiangsu
	25	*Styphnolobium japonicum* (L.) Schott	Suqian, Jiangsu
	26	*Amorpha fruticosa* L.	Suqian, Jiangsu
	27	*Erythrina crista-galli* L.	Wenjiang, Sichuan
Elaeagnaceae	28	*Hippophae rhamnoides* L. ‘Shenqiuhong’	Huai’an, Hebei
	29	*Hippophae rhamnoides* L. ‘Wucifeng’	Huai’an, Hebei
	30	wild *Hippophae rhamnoides* subsp. *sinensis* Rousi	Huai’an, Hebei
	31	cultural *Hippophae rhamnoides* subsp. *sinensis* Rousi	Tieling, Liaoning
	32	*Hippophae rhamnoides* L. ‘Zhuangyuanhuang’	Wenchuan, Sichuan
	33	*Elaeagnus lanceolate* Warb. apud Diels	Wenchuan, Sichuan
Taxaceae	34	*Taxus cuspidata* Siebold & Zucc.	Suqian, Jiangsu
	35	*Taxus* × *media* Rehder	Ya’an, Sichuan
	36	*Taxus wallichiana* var. *mairei* (Lemee & H. Léveillé) L. K. Fu & N. Li	Ya’an, Sichuan
	37	*Taxus wallichiana* Zucc.	Xichuang, Sichuan
	38	*Taxus yunnanensis* W.C. Cheng & L.K. Fu	Ya’an, Sichuan
Juglandaceae	39	*Juglans regia* L. ‘Chuanzao1’	Qingbaijiang, Sichuan
	40	*Juglans sigillata* Dode ‘Maerkang’	Maerkang, Sichuan
	41	*Juglans sigillata* Dode ‘Muzhilinhe’	Gulin, Sichuan
	42	*Juglans regia* L. ‘Yanyuanzao’	Yanyuan, Sichuan
Oleaceae	43	*Fraxinus pennsylvanica* Marsh.	Wenjiang, Sichuan
	44	*Syringa oblata* Lindl.	Wenjiang, Sichuan
	45	*Ligustrum* × *vicaryi* Rehder	Wenjiang, Sichuan
	46	*Ligustrum lucidum* Ait.	Wenjiang, Sichuan
Asparagaceae	47	*Polygonatum cyrtonema* Hua	Wenchuan, Sichuan
	48	*Polygonatum sibiricum* Delar. ex Redoute	Dujiangyan, Sichuan
Berberidaceae	49	*Berberis diaphana* Maxim.	Wenchuan, Sichuan
	50	*Berberis soulieana* Schneid.	Wenchuan, Sichuan
Lauraceae	51	*Litsea baviensis* Lec.	Jinniu, Sichuan
	52	*Litsea elongate* (Wall. ex Nees) Benth. et Hook. f.	Jinniu, Sichuan
Malvaceae	53	*Firmiana simplex* (L.) W. Wight	Wenjiang, Sichuan
	54	*Hibiscus mutabilis* L.	Jinniu, Sichuan
Euphorbiaceae	55	*Croton tiglium* L.	Ya’an, Sichuan
	56	*Triadica sebifera* (Linnaeus) Small	Jiangyou, Sichuan
Cupressaceae	57	*Cryptomeria japonica* (L.f.) D. Don	Wenjiang, Sichuan
Aquifoliaceae	58	*Ilex chinensis* Sims	Dujiangyan, Sichuan
Calycanthaceae	59	*Chimonanthus campanulatus* R.H. Chang & C.S. Ding	Jinniu, Sichuan
Fagaceae	60	*Quercus semecarpifolia* Smith	Wenchuan, Sichuan
Poaceae	61	*Triticum aestivum* L.	Wenjiang, Sichuan
Podocarpaceae	62	*Podocarpus macrophyllus* (Thunb.) Sweet	Wenjiang, Sichuan
Salicaceae	63	*Idesia polycarpa* Maxim.	Wenjiang, Sichuan
Sapindaceae	64	*Koelreuteria bipinnata* Franch.	Wenjiang, Sichuan

**Table 2 genes-15-00647-t002:** Chromosome number and length of the 42 species used in this work.

Accession	Species	Chromosome Number	Chromosome Length	Karyotype Asymmetry
A1	*Cercis chinensis*	2n = 14	1.58–2.21 μm	1.40
A2	*Piptanthus nepalensis*	2n = 18	1.65–3.02 μm	1.83
A3	*Robinia pseudoacacia*	2n = 22	1.37–1.68 μm	1.23
A4	*Robinia pseudoacacia* f. *decaisneana*	2n = 22	0.77–1.40 μm	1.82
A5	*Robinia pseudoacacia* ‘Idaho’	2n = 22	1.09–1.47 μm	0.74
A6	*Styphnolobium japonicum*	2n = 28	0.98–3.19 μm	3.26
A7	*Amorpha fruticosa*	2n = 40	1.44–2.18 μm	1.51
A8	*Erythrina crista-galli*	2n = 42	1.37–3.32 μm	2.42
A9	*Polygonatum cyrtonema*	2n = 18	1.54–4.49 μm	2.92
A10 *	*Polygonatum sibiricum*	2n = 18	1.47–6.77 μm	4.61
A11	*Croton tiglium*	2n = 20	1.65–3.61 μm	2.19
A12	*Chimonanthus campanulatus*	2n = 22	1.33–1.93 μm	1.45
A13	*Cryptomeria japonica*	2n = 22	3.09–6.14 μm	1.99
A14	*Quercus semecarpifolia*	2n = 24	1.47–3.10 μm	3.11
A15	*Litsea baviensis*	2n = 24	1.86–4.04 μm	2.17
A16	*Litsea elongate*	2n = 24	1.58–4.21 μm	2.66
B1	*Taxus yunnanensis*	2n = 24	2.46–6.28 μm	2.55
B2	*Taxus* × *media*	2n = 24	2.39–6.88 μm	2.88
B3	*Taxus cuspidata*	2n = 24	2.60–6.32 μm	2.43
B4	*Taxus wallichiana*	2n = 24	2.42–6.42 μm	2.65
B5	*Taxus wallichiana* var. *mairei*	2n = 24	2.81–6.67 μm	2.37
B6	*Hippophae rhamnoides* ‘Wucifeng’	2n = 24	1.40–2.60 μm	1.86
B7	*Hippophae rhamnoides* ‘Shenqiuhong’	2n = 24	1.26–2.46 μm	1.95
B8	*Hippophae rhamnoides* ‘Zhuangyuanhuang’	2n = 24	1.44–2.81 μm	1.95
B9	cultural *Hippophae rhamnoides* subsp. *sinensis*	2n = 24	1.30–2.18 μm	1.68
B10	wild *Hippophae rhamnoides* subsp. *sinensis*	2n = 24	1.05–2.32 μm	2.21
B11	*Elaeagnus lanceolate*	2n = 28	1.02–3.68 μm	3.61
B12	*Berberis diaphana*	2n = 28	1.54–2.84 μm	1.84
B13	*Berberis soulieana*	2n = 28	2.18–3.51 μm	1.61
B14	*Koelreuteria bipinnata*	2n = 32	0.77–1.82 μm	2.36
B15	*Juglans regia* ‘Chuanzao1’	2n = 34	0.70–1.65 μm	2.36
B16	*Juglans sigillata* ‘Maerkang’	2n = 34	0.63–1.61 μm	2.56
C1	*Juglans sigillata* ‘Muzhilinhe’	2n = 34	0.95–2.28 μm	2.40
C2	*Juglans regia* ‘Yanyuanzao’	2n = 34	0.91–2.19 μm	2.41
C3	*Bletilla formosana* ‘Leshan’	2n = 34	0.98–3.68 μm	3.76
C4	*Bletilla striata* f. ‘Dujiangyan’	2n = 34	0.88–3.96 μm	4.50
C5	*Bletilla striata* f. ‘Wenchuan’	2n = 34	1.33–4.74 μm	3.56
C6	*Bletilla striata* f. ‘Neijiang’	2n = 34	1.40–3.16 μm	2.26
C7	*Bletilla striata* f. ‘Baoshan’	2n = 34	1.26–3.61 μm	2.87
C8	*Bletilla striata* f. ‘Nanan’	2n = 34	1.12–3.86 μm	3.45
C9	*Bletilla striata* f. ‘Honghe’	2n = 34	1.23–4.04 μm	3.28
C10	*Bletilla ochracea* ‘Zigong’	2n = 34	1.12–2.63 μm	2.35
C11	*Bletilla ochracea* ‘Dujiangyan’	2n = 34	1.19–3.02 μm	2.63
C12	*Firmiana simplex*	2n = 36	0.98–2.21 μm	2.26
C13	*Hibiscus mutabilis*	2n = 90	1.09–2.98 μm	2.73
C14	*Podocarpus macrophyllus*	2n = 36	1.93–5.26 μm	2.73
C15	*Idesia polycarpa*	2n = 38	0.77–2.46 μm	3.19
C16 *	*Ilex chinensis*	2n = 40	0.63–1.58 μm	2.51
D1	*Triticum aestivum*	2n = 42	3.85–6.88 μm	1.79
D2	*Fraxinus pennsylvanica*	2n = 46	0.88–1.96 μm	2.23
D3	*Syringa oblata*	2n = 46	0.98–1.93 μm	1.97
D4	*Ligustrum* × *vicaryi*	2n = 46	0.95–2.14 μm	2.25
D5	*Ligustrum lucidum*	2n = 46	0.70–2.11 μm	3.01
D6 *	*Triadica sebifera*	2n = 88	0.60–1.23 μm	2.05
D7	*Zanthoxylum nitidum*	2n = 66	1.40–2.56 μm	1.83
D8	*Zanthoxylum bungeanum* ‘Hanyuanhuajiao’ 1	2n = 76	0.84–1.72 μm	2.05
D9	*Zanthoxylum bungeanum* ‘Hanyuanhuajiao’ 2	2n = 134	0.88–1.82 μm	2.07
D10	*Zanthoxylum bungeanum* ‘Hanyuanhuajiao’ 3	2n = 136	0.91–2.14 μm	2.35
D11	*Zanthoxylum bungeanum* ‘Hanyuanhuajiao’ 4	2n = 160	0.74–1.68 μm	2.27
D12	*Zanthoxylum armatum* ‘Hongyatengjiao’ 1	2n = 96	0.70–1.47 μm	3.01
D13	*Zanthoxylum armatum* ‘Tengjiao’	2n = 100	0.88–2.00 μm	2.27
D14	*Zanthoxylum armatum* ‘Hongyatengjiao’ 2	2n = 102	0.95–2.04 μm	2.14
D15	*Zanthoxylum armatum* ‘Jiuyeqing’	2n = 104	0.88–2.00 μm	2.27
D16	*Zanthoxylum armatum* ‘Putaoqingjiao’	2n = 132	0.74–1.79 μm	2.42

Note: asterisk (*) in Table 2 indicates chromosome number of three species are first reported.

## Data Availability

The original contributions presented in the study are included in the article/Appendix A, further inquiries can be directed to the corresponding author.

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
