# Peer review of "Karyotype Description and Comparative Chromosomal Mapping of 5S rDNA in 42 Species"

_genes, 2024, doi:10.3390/genes15050647_

Round 1

Reviewer 1 Report

Comments and Suggestions for Authors

This paper describes the 5S rDNA locations in  21 plant species, some are novel for this study, but others have been reported before (e.g. in Triticum aestivum ). There are excellent FISH images and interesting analysis, but as the paper is written, I can not recommend it in its current form.

Overall, the English is very poor with many grammatical mistakes, and incorrect words used probably caused by automated translation. In many cases the meaning cannot be understood. For example ‘Summarily [!], 5S rDNA has been confirmed to be a significant cytogenetic symbol [!] by tandem formatting [!] and giving in multicopy [!] numbers with unusual chromosomal allocation [!].  Converged [?] seeds were germinated in Petri dishes with moistened filter paper and stayed at 25℃ in the daytime and at 18℃ in the night until the roots touched [!] ~2 cm in length.’ ; ‘Raw data were executed [!] using the Photoshop …’

The abstract is quite confusing with analysis of 21 species  ‘checked in which 5S rDNA has not yet been explored’, but later it says  ‘19 species have been reported for the first time?’ There is also the mention of 37, 33 and 64 plants respectively -   To me this does not make sense although I think there were several plants from the same species. Also I find the conclusion ‘The potential origin of signal pattern diversity was probably caused by chromosome rearrangement, polyploidization, self-incompatibility, and chromosome satellites.’ too general and a mixture of plant reproduction and chromosome biology mechanisms.

The introduction and other parts of the manuscript have many lists of species and references some of which would be better represented in tables, but only if this is relevant for the present paper. There is a supplementary table, but those are additional to the list in the introduction as far as I can see.

I understand that the current analysis and relationship trees only use the 64 plants of this study, but no results from previous published work is used although the literature had been extensively surveyed.

I also found factual misunderstandings. Most 5S rDNA units have a very conserved 120-150bp transcribed region  and variable large (300-600bp) spacer that can be species or locus specific. Probers for FISH normally will include the whole unit or parts of the conserved region.  I am not sure it is relevant to list all probes used in the literature.

The paragraphs 3.3. where you propose the origin of 5S rDNA signal diversity, has no references to your arguments as evidence for causes such as transposable elements or chromosome rearrangements etc.  These are then given in the discussion section 4.3. I suggest to delete the section 3.3.

Please delete the first paragraph of the discussion that is left over from the author instructions. The remaining of the discussions brings together some interesting thoughts and discussion, but parts are too lengthy and go too far away from the purpose of the study you conducted and that is not designed to answer any questions that are raised by the literature you cite.

Comments on the Quality of English Language

Overall, the English is very poor with many grammatical mistakes, and incorrect words used probably caused by automated translation. In many cases the meaning cannot be understood. For example ‘Summarily [!], 5S rDNA has been confirmed to be a significant cytogenetic symbol [!] by tandem formatting [!] and giving in multicopy [!] numbers with unusual chromosomal allocation [!].  Converged [?] seeds were germinated in Petri dishes with moistened filter paper and stayed at 25℃ in the daytime and at 18℃ in the night until the roots touched [!] ~2 cm in length.’ ; ‘Raw data were executed [!] using the Photoshop …’

There are many more such strange and confusing sentences. 

Author Response

Dear Editor,

We have revised our manuscript based on the suggestions from the Editor and Reviewer 1. We have attached the point-by-point response. Please see the attachment. 

Wishes

Xiaomei

Reviewer 2 Report

Comments and Suggestions for Authors

In this study, the authors investigated the 5S rDNA site number, position, and origin of signal pattern diversity in 21 plants species. The study falls into the scope of Genes and thus could be published. Before this can be done, a major revision should be made as specified below.

First, the English must be revised. Some parts of the text are hard to comprehend, not only grammatically, but also structurally. 

In several parts of the text the authors have long phrases describing what was found in some species. These parts are not pleasant to read.

Introduction

-              Second paragraph is unnecessary to this study.

-              Third and fourth paragraphs starts with the same word (Conversely), please edit one of them.

-              Third paragraph is very tiring. Perhaps it would be better to organize this information on a table.

-              I do not understand what the authors means in some parts of the text. For instance, line 66, “FISH signal position was diverse and abundant”. I understand that the FISH signal position was diverse, but I do not understand what abundant means. I think it refers to the numbers of chromosomes with FISH signals, but it was not addressed in this paragraph. In some parts of results and discussion the authors also used the word abundant. I understood that this means signal intensity, but it can be a variation in the FISH experiments or chromosome preparation. Please check!

-              Line 82, what do you mean with “lacked discrimination”?

-              The authors should compare in this section and discussion section the 5S rDNA organization in animal species, such as birds, fish, and mammals. That could increase the interest of readers who work with species other than plants.

Material and Methods

-              I was confused about the number of species that the authors investigated. I suggest to used only the number of species investigated (21 species) and not the in the number of plants (64), as used in some parts of the text.

-              Lines 109-110. Change “karyotype realignments” to “karyotype reshuffling” or “karyotype reorganization”. Please check all the section.

Discussion

-              Lines 327-330 must be deleted!!!

-              Line 339. Remove “in this study”.

-              Line 341. Use italic to the genus Ilex.

-              Line 360. What is the meaning of “(L.) L.” and “Ten” in this phrase?

-              Line 360 the authors said that intraspecific chromosome number variation has been found in species of Cuscuta epithymum and Cuscuta planiflora. At line 365 they included the genus Zanthoxylum. This genus should be cited in the beginning of the paragraph as well.

-              Line 372, what do you mean with reversal?

Comments on the Quality of English Language

Extensive editing of English language required.

Author Response

Dear Editor,

We have revised our manuscript based on the suggestions from the Editor and Reviewer 2. We have attached the point-by-point response. Please see the attachment. 

Wishes

Xiaomei

Round 2

Reviewer 2 Report

Comments and Suggestions for Authors

This is a revised manuscript.

First, the English must be revised not only grammatically, but structurally.

In my opinion, the second, third, fourth, fifth and sixth paragraphs of the introduction are too boring, with so many examples. I suggest to the authors to try to improve these paragraphs to make them more attractive.

Minor issues:

Lines 21-22: Explain how the 5S rDNA signal can be found outside of chromosomes. Also check the lines 277-283. I do not understand what the authors means.

Line 309: There are two dots in the end of this paragraph.

Comments on the Quality of English Language

The English of the manuscript must be revised not only grammatically, but structurally. 
